# Integrating sex-bias into studies of archaic introgression on chromosome X

**Elizabeth T. Chevy**[1]*, **Emilia Huerta-Sánchez**[1,2], **Sohini Ramachandran**[1,2,3]*

**1** Center for Computational Molecular Biology, Brown University, Providence, Rhode Island, United States of America, **2** Department of Ecology, Evolution, and Organismal Biology, Brown University, Providence, Rhode Island, United States of America, **3** Data Science Initiative, Brown University, Providence, Rhode Island, United States of America

☉ These authors contributed equally to this work.

* elizabeth_chevy@brown.edu (ETC); sohini_ramachandran@brown.edu (SR)

## Abstract

Evidence of interbreeding between archaic hominins and humans comes from methods that infer the locations of segments of archaic haplotypes, or 'archaic coverage' using the genomes of people living today. As more estimates of archaic coverage have emerged, it has become clear that most of this coverage is found on the autosomes— very little is retained on chromosome X. Here, we summarize published estimates of archaic coverage on autosomes and chromosome X from extant human samples. We find on average 7 times more archaic coverage on autosomes than chromosome X, and identify broad continental patterns in this ratio: greatest in European samples, and least in South Asian samples. We also perform extensive simulation studies to investigate how the amount of archaic coverage, lengths of coverage, and rates of purging of archaic coverage are affected by sex-bias caused by an unequal sex ratio within the archaic introgressors. Our results generally confirm that, with increasing male sex-bias, less archaic coverage is retained on chromosome X. Ours is the first study to explicitly model such sex-bias and its potential role in creating the dearth of archaic coverage on chromosome X.

**Data Availability Statement:** All code and data necessary to replicate simulations and plot main figures is available in a public GitHub repository (https://github.com/ramachandran-lab/archaic-chrX-sexbias).

## Author summary

Tens of thousands of years ago, humans interbred with our close hominin relatives (*e.g.* Neanderthals), which we know from finding segments of archaic hominin DNA in our genomes. Up to 4% of a human genome may be archaic DNA, but most of that archaic part is on the autosomes (the non-sex chromosomes). Chromosome X usually contains 3 to 10 times less archaic DNA than the autosomes. Also unlike the autosomes, it is always passed down by mothers, but only sometimes by fathers. There are several hypotheses for why chromosome X has less archaic DNA than the autosomes; one that has not been fully explored is whether the archaic hominins that interbred with our ancestors were mostly male or mostly female, known as 'sex-bias'. In this paper, we use simulations to investigate whether sex-bias could produce less archaic DNA on chromosome X. Using simulation studies, we find that when the archaics are mostly male, modern humans end up with less

**Funding:** This research was supported by US National Institutes of Health (NIH) grant R01 GM118652, NIH grant R35 GM139628, and National Science Foundation (NSF) CAREER award DBI-1452622, and support from the Erling-Persson Family Foundation and the Knut and Alice Wallenberg Foundation to SR, as well as NIH grant R35 GM128946, and an Alfred P. Sloan Research Fellowship to EH-S. ETC was a trainee supported under the Brown University Predoctoral Training Program in Biological Data Science (NIH T32 GM128596). Any opinions, findings, and conclusions or recommendations expressed in this material are those of the author(s) and do not necessarily reflect the views of any of the funders. The funders had no role in study design, data collection and analysis, decision to publish, or preparation of the manuscript.

**Competing interests:** The authors have declared that no competing interests exist.

archaic DNA on chromosome X than their autosomes, compared to when there is a female-bias or no sex-bias. Therefore, male sex-bias could be contributing to the difference in the amount of archaic DNA on chromosome X versus the autosomes. Of course, there are still plenty of other factors to be explored about how demography and selection have shaped our DNA. Studying patterns like this helps us learn more about early hominin natural history, and contextualizes archaic interbreeding events among other sex-biased events in human history.

## Introduction

Secondary contact between archaic hominin (*e.g.*, Neanderthal) and modern human groups around 50,000 years before present generated individuals with genomes containing a mixture of both archaic and modern human genetic material, and genomic evidence of this archaic genetic material survives in the human gene pool today. Recent computational methods [1–3] enable inferring which 'tracts' (or, segments of haplotypes) of an individual's genome are retained from an archaic introgression event. We refer to the proportion of an individual's genome, or of a given chromosome of interest, contained within archaic tracts as that individual's 'archaic coverage'. Among extant modern humans who harbor some archaic genetic material, archaic coverage across the autosomes typically ranges between 1–3% [3]. However, as Fig 1 demonstrates, the archaic coverage observed on chromosome X is much less than that observed on the autosomes [1–5].

Despite the scientific attention paid to autosomal archaic coverage in modern humans, the precise factors behind the archaic coverage discrepancy between chromosome X and autosomes remain unresolved. Many studies have examined how different modes of selection and introgression may have reduced archaic coverage to the levels observed in modern humans living today [1, 2, 6–11]. Of these, some [1, 2, 6, 8] have attempted to study how evolutionary processes influence archaic coverage on chromosome X. Sankararaman *et al.* [1] posited that the large-X effect, an excess of hybrid incompatibility loci on chromosome X, reduced male fitness in admixed individuals and consequently drove archaic variants from these X-linked loci. Still fewer studies [2, 6, 8] have modeled the unique inheritance pattern of chromosome X, thereby acknowledging that X-linked mutations can be differentially exposed to selection in males versus females. These studies have argued that background selection against archaic variants is sufficient to explain the depletion of chromosome X archaic coverage.

Sex-bias—a phenomenon that is readily observed in human societies today [12–14], as well as in historical introgression events [15, 16]—could also drive the observed discrepancy in archaic coverage between chromosome X and the autosomes. However, to our knowledge, no study has yet considered how sex-biased demography (unequal representation of males and females in demographic processes such as migration or population founding events) might interact with sex-specific chromosome X inheritance and natural selection to produce the paucity of archaic coverage observed today on chromosome X. For example, male-bias in the archaic hominin donor pool could result in lower archaic coverage on chromosome X in modern humans today. If male archaics disproportionately contributed deleterious archaic variation, recessive variants would be more efficiently purged from chromosome X than the autosomes, due to the hemizygous state of chromosome X in males. Even for neutral archaic variation, a disproportionate contribution from males would leave less archaic coverage on chromosome X due to the lower copy number of chromosome X in males than females.

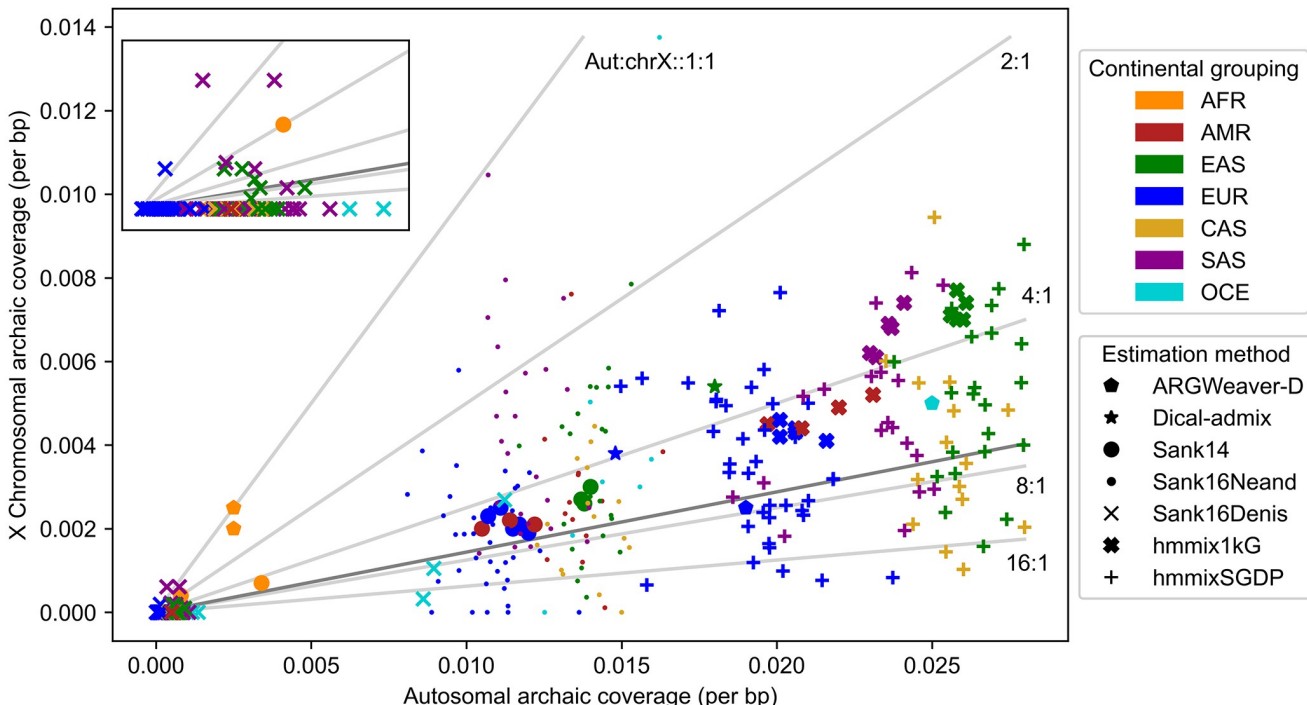

**Fig 1. Summary of published empirical estimates of archaic coverage in global groups show greater coverage rates on autosomes than chromosome X.** Each point indicates the per-bp rate of archaic coverage on autosomes and chromosome X in a global human group; see S1 Table for numerical values. Colors indicate continental group; see Table 1 for key. Shapes indicate estimation method; see S2 Table for details. Light grey clines show autosomal to chromosome X coverage ratios of 1 (*i.e.*, equal per bp coverage), 2 (*i.e.*, autosomal archaic coverage is double that of chromosome X), 4, 8, and 16. Dark grey cline shows median autosomal to chromosome X coverage ratio of 6.9. Nearly all points are well below the 1:1 line, indicating greater per-bp coverage on autosomes than chromosome X.

In this study, we test whether sex-bias in archaic-human introgression events may have contributed to relatively low archaic coverage on chromosome X versus autosomes in modern humans, and how sex-bias can interact with dominance to produce the observed discrepancy in archaic coverage. To do so, we simulate an archaic introgression scenario using either chromosome X or autosomal inheritance using SLiM3 [17]. We calculate the archaic coverage observed in simulated haplotypes at the present day (end of the simulation), and investigate how dominance and sex-bias affect the amount, distribution, and temporal dynamics of archaic coverage on chromosome X.

## Results

### Chromosome X harbors little archaic coverage in humans today

We first review previously published estimates of archaic coverage on chromosome X and the autosomes in present-day human groups (Fig 1; [1–5]). In all but one of the groups studied, chromosome X retains less archaic coverage than the autosomes. Only a Mandenka sample group from the 1000 Genomes Project (1kG) [18] has equal rates of archaic coverage on chromosome X and autosomes.

The autosomal to chromosome X coverage ratio (aut:chrX) reflects how many times greater the per-base pair rate of archaic coverage is on the autosomes than on the chromosome X. The median aut:chrX ratio considering all estimates is 6.9; excluding estimates specifically of Denisovan coverage [4], the median is 5.2.

**Table 1. Population abbreviations used in Figs 1 and 2.**

|  | | Superpopulation | | | Population |
|---|---|---|---|---|---|
| AFR | | African | | | |
| AMR | | American | | | |
| | | | CLM | | Colombian in Medellin, Colombia |
| | | | MXL | | Mexican Ancestry in Los Angeles, CA |
| | | | PEL | | Peruvian in Lima, Peru |
| | | | PUR | | Puerto Rican in Puerto Rico |
| EAS | | East Asian | | | |
| | | | CDX | | Chinese Dai in Xishuangbanna, China |
| | | | CHB | | Han Chinese in Beijing, China |
| | | | CHS | | Han Chinese South |
| | | | JPT | | Japanese in Tokyo, Japan |
| EUR | | European | | | |
| | | | FIN | | Finnish in Finland |
| | | | GBR | | British in England and Scotland |
| | | | IBS | | Iberian populations in Spain |
| | | | TSI | | Toscani in Italia |
| CAS | | Central Asian / Siberian | | | |
| SAS | | South Asian | | | |
| | | | BEB | | Bengali in Bangladesh |
| | | | GIH | | Gujarati Indian in Houston, TX |
| | | | ITU | | Indian Telugu in the UK |
| | | | PJL | | Punjabi in Lahore, Pakistan |
| | | | STU | | Sri Lankan Tamil in the UK |
| OCE | | Oceanian | | | |

Population classifications and codes are defined by the 1000 Genomes Project [18].

Most estimates of archaic coverage displayed in Fig 1 are designed to identify coverage donated from Neanderthal specifically (ARGWeaver-D [3], Dical-admix [2], and Sankararaman *et al.*'s CRF [1, 4]), while CRF results from Sankararaman *et al.* [4] also provide estimates specific to Denisovan ancestry (S2 Table). The method of Skov *et al.* [5] (hmmix) results in larger amounts of archaic coverage (compared to the other methods) likely because this method does not use a specific archaic donor to infer archaic ancestry. However, ratios of the autosome to chromosome X coverage rates computed from introgression maps inferred using hmmix are comparable to the other methods.

Of the 120 human groups in which Denisovan coverage was identified, only 15 showed evidence of retaining any Denisovan coverage at all on chromosome X. Groups with Denisovan coverage on chromosome X were mostly found in Oceania and East Asia (S1 Table).

## Archaic coverage retention differs among 1kG sample groups

While chromosome X coverage rates are lower than autosomal rates in groups around the globe, we observe some broad patterns among continental groups. We concentrate on coverage tracts identified by hmmix in 1kG data, as run by the authors of this study (Fig 2; see Methods).

In accordance with previous studies, we see the greatest mean autosomal coverage rate in the EAS 1kG continental group (2.6%), and least in the EUR 1kG continental group(2.1%).

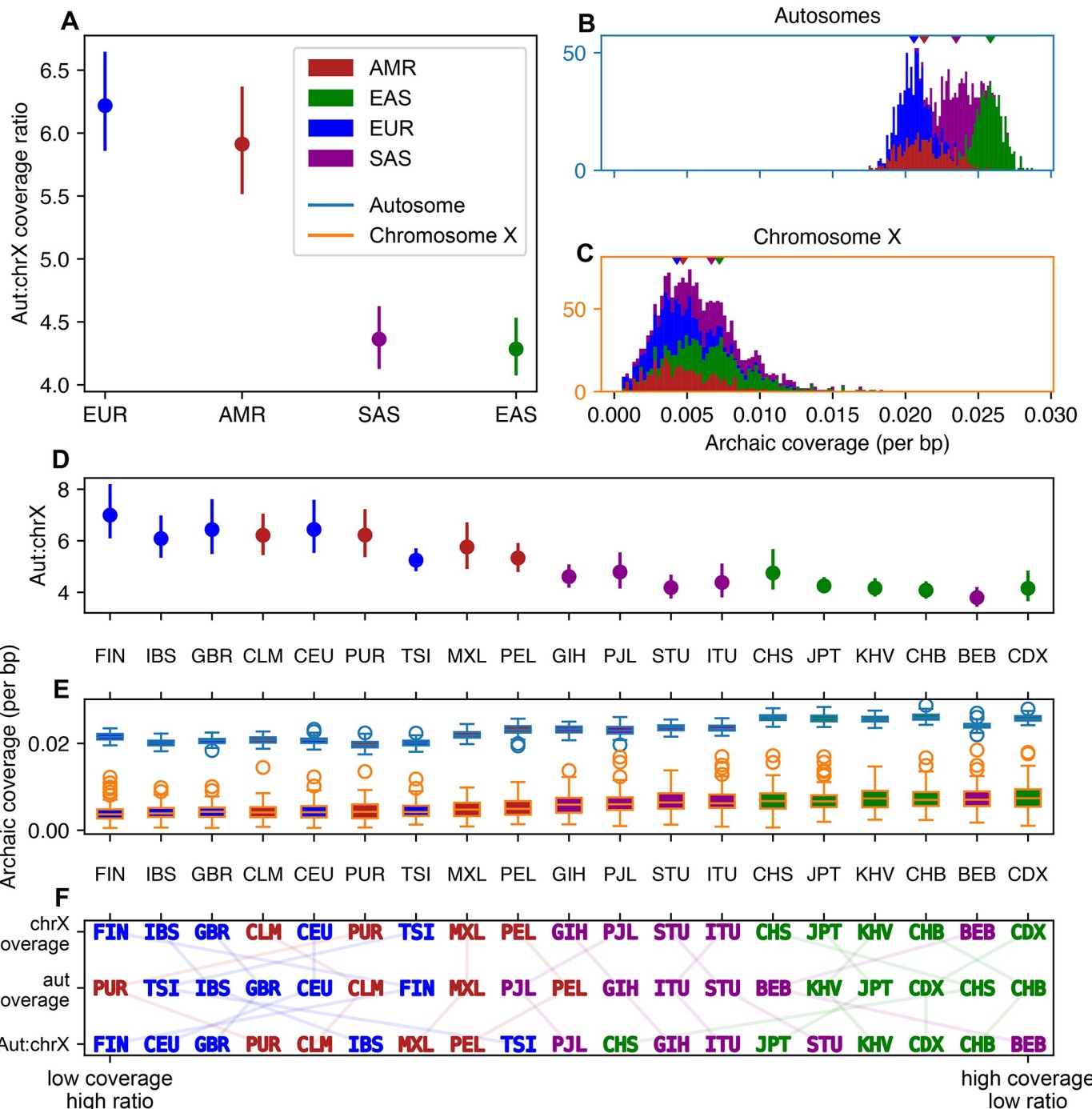

**Fig 2. Distribution of archaic coverage between autosomes and chromosome X amongst 1kG samples.** Coverage estimated with `hmmix` [5] using data from the 1000 Genomes Project [18]. Chromosome X coverage tracts have been filtered to remove pseudoautosomal regions and putative targets of selection (75% confidence regions from S2 Table of Skov *et al.* [19]). See Table 1 for key to abbreviations. **A** Mean aut:chrX archaic coverage ratio among continental groups. Error bars show bootstrapped 95% confidence intervals around the means. **B** Distributions of archaic coverage on autosomes and **C** chromosome X across individuals, colored by 1kG continental group. Carets indicate means. **D** Summaries of coverage ratios, as well as **E** coverage distributions on the two chromosome types among 1kG sample groups. 1kG sample groups are sorted by ascending archaic coverage amounts on chromosome X. **F** Rank orderings of 1kG sample groups by mean chromosome X coverage, mean autosomal coverage, and aut:chrX ratio, sorted from least to greatest.

Mean chromosome X coverage rate is also greatest in EAS and least in EUR (0.7% and 0.4%; Fig 2B and 2C).

At the continental group level, we find significantly different mean autosomal and chromosome X coverage rates among all choices of 1kG continental group(all pairwise $p$-values $\leq$ 0.0051 < 0.0083 Bonferroni-corrected 5% level for Welch's $t$-test). We also see the same ordering of 1kG continental groups across autosomes and chromosome X for coverage amounts: EAS > SAS > AMR > EUR.

While coverage ordering does not imply a particular ratio ordering, we do observe that mean aut:chrX ratio for the 1kG continental groups follows the reverse order as coverage. EUR and AMR have significantly higher aut:chrX ratios than SAS and EAS: combined EUR/AMR aut:chrX ratio is 6.1, and SAS/EAS is 4.3 (Fig 2A).

Broad continental patterns among 1kG continental groups are generally preserved when considering specific 1kG sample groups (Fig 2D and 2E). Although, unlike the 1kG continental groups, ordering by each of the three metrics (mean chromosome X coverage, autosomal coverage, and aut:chrX ratio) produce different 1kG sample group-level rankings (Fig 2F). No 1kG sample group pairs have significantly different mean aut:chrX ratios across the ((EUR, AMR), (EAS, SAS)) boundary. For chromosome X coverage rates, the only significantly different 1kG sample group pairs within (EUR, AMR) or (EAS, SAS) are FIN/PEL and GIH/CDX. Most pairs (147/171) of 1kG sample groups have significantly different autosomal coverage rates, with the notable exception of within EAS. KHV/CHB are the only pair of EAS 1kG sample groups where autosomal coverage differs. The broad coherence in coverage among 1kG sample groups within each 1kG continental group suggests that continental group-level histories of admixture may explain these patterns. (See S7 Fig and Discussion).

## Dominance, sex-bias affect purging of archaic coverage from chromosome X

In order to study genomic patterns of archaic coverage, we implemented demographic simulations of a single pulse of introgression from a donor archaic population to the recipient modern human population (S1 Fig). Our simulation model closely follows that of Kim *et al.* [8]; complete simulation details are provided in Methods. For each simulation, we set the dominance model of mutations (all "additive" with $h = 0.5$, or all "recessive" with $h = 0.0$), and the sex-ratio of the introgressors ($p$, fraction of the introgressing individuals that are female). All mutations are deleterious, with most weakly so ($\approx$ 74% of selection coefficients $s \leq 0.01$). We generated autosomal and X-chromosomal haplotypes representing extant modern humans under the appropriate model of inheritance for each chromosome type. For each modern human haplotype sampled at the end of our simulation, we calculate its archaic coverage as the proportion of bases identical by descent to any of the introgressors, and contained within a contiguous tract of at least 500 bp.

Our simulations replicate the empirical observation of less archaic coverage on chromosome X than autosomes (Fig 3). Regardless of dominance model or sex-bias, the archaic coverage ratio between autosomes and chromosome X is greater than 1, indicating more autosomal than chromosome X archaic coverage.

We also observe that archaic coverage is less than the initial introgression fraction of 5% in all models (Fig 3). Were archaic variants neutral in the modern human population, we would expect 5% archaic coverage to be maintained in the recipient population sample. However, the effective size of the archaic population in our simulations is smaller than that of the recipient modern human population. Deleterious variants that drifted to high frequency in the smaller

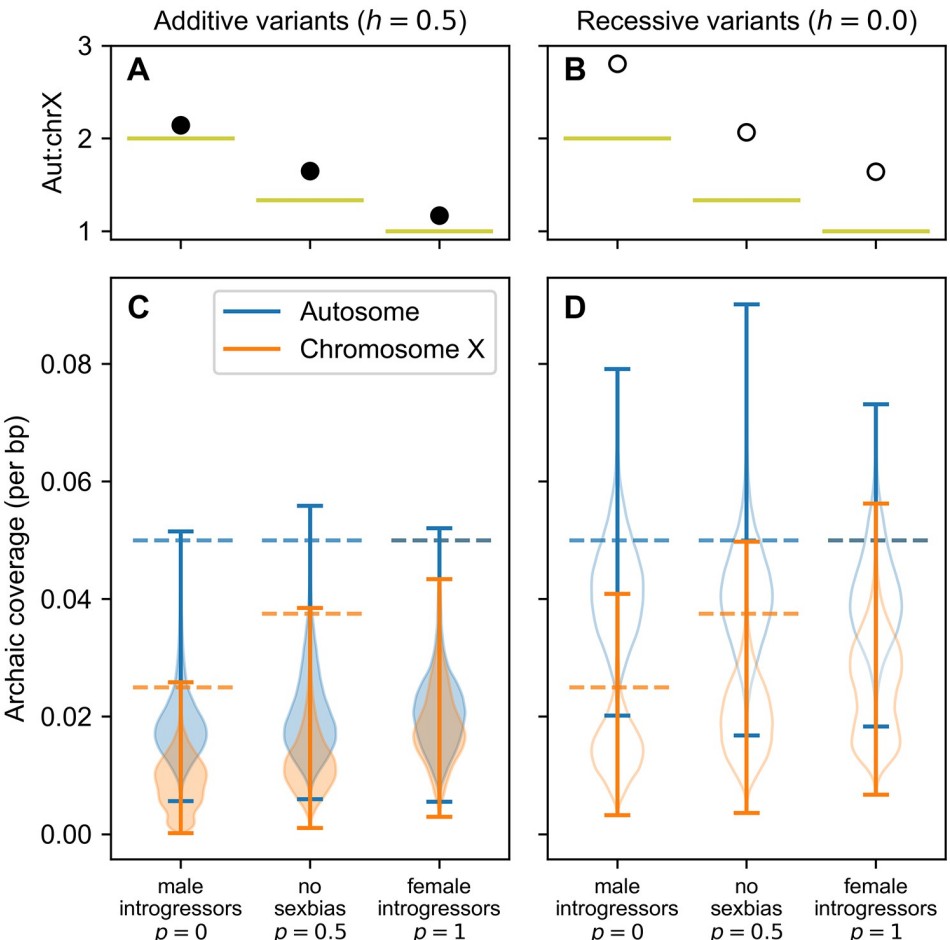

**Fig 3. The effects of dominance and sex-bias on chromosome X archaic coverage. A, B** Mean coverage ratios between a simulated autosome and chromosome X, estimated from 10,000 pairwise combinations of simulated autosomal and X chromosomes (see Methods). Bootstrapped 95% confidence intervals around these means are not visible because they are smaller than the markers. Filled points reflect additive variants ($h = 0.5$); hollow points reflect recessive variants ($h = 0.0$). Horizontal lines denote autosome to chromosome X ratio of archaic haplotypes in initial introgression pulse. See Eq 1. **C, D** Filled violins depict additive variants; hollow violins depict recessive variants. Dashed horizontal lines denote magnitude of initial introgression pulse.

archaic population are subsequently selected against after introgression into the larger modern human population, thereby 'purging' some of the initial introgressed sequences from the extant haplotypes. These results are consistent with previous studies of archaic purging by background selection [6, 7].

Under both dominance models, we observe a positive correlation between the female fraction of the introgressors and the rate of archaic coverage on chromosome X (Additive model: Pearson's $\rho = 0.53$, $p < 10^{-20}$; Recessive model: Pearson's $\rho = 0.37$, $p < 10^{-20}$; Fig 3). Autosomes are not inherited in a sex-specific fashion, so, as expected, we see a weaker relationship between archaic coverage and sex-bias on the autosomes (Additive model: Pearson's $\rho = 0.10$, $p < 10^{-20}$; Recessive model: Pearson's $\rho = -0.02$, $p < 10^{-18}$).

Introgression is an individual-level process. Therefore, the number of introgressing archaic haplotypes differs between autosomes and chromosome X depending on the sexes of the

introgressors. We expect the ratio of autosomal haplotypes ($H^A$) to X chromosomes ($H^X$) to follow

$$\frac{H^A}{H^X} = \frac{2}{1+p} \tag{1}$$

for an introgressing class with female fraction $p$ (depicted as chartreuse lines in Fig 3A and 3B, and S2 Fig). For example, both the quantity and configuration of introgressing archaic haplotypes are identical between inheritance modes when all introgressors are female ($p = 1$; *i.e.* all chromosome X haplotypes introgress in pairs, just like autosomes). Indeed, we observe the smallest autosome to chromosome X coverage difference under the additive model when all introgressors are female (autosomal mean 0.0211 vs. chromosome X mean 0.0181; Fig 3C).

Thus, we both expect and observe a negative correlation between female fraction and the archaic coverage ratio between autosomes and chromosome X (Fig 3A and 3C, and S2 Fig). These findings are driven solely by the sex ratio of the introgressors and the inheritance model of the chromosome types, as our simulations do not invoke additional mechanisms such as chromosome-specific distributions of fitness effects, or hybrid incompatibilities.

However, not all of the difference between autosomal and chromosome X coverage can be attributed to the initial haplotype ratio within the introgression pulse. We observe autosome to chromosome X coverage ratios much higher than predicted by Eq 1 in Fig 3A and 3C, particularly when $p < 0.6$ (S2 Fig). This effect is exacerbated when the sex ratio of the recipient population is male-biased; see for example S3 Fig).

The model of recessive variants produces greater coverage differences between autosomes and chromosome X than the model of additive variants across all degrees of sex-bias (Fig 3 and S2 Fig). This effect is driven by the heterotic advantage of archaic coverage in paired chromosomes when variants are recessive.

## Length distribution of archaic coverage tracts differs between chromosome X and autosome

Although chromosome X harbors less archaic coverage than autosomes, its coverage is contained within longer tracts than autosomal coverage (S4 Fig). As expected, due to the reduced frequency of recombination events in chromosome X inheritance and the preferential purging of archaic coverage from chromosome X, there are fewer total archaic coverage tracts on chromosome X than an autosome (S4(A) Fig). Compared to a simple model exponential model of coverage tract length decay from introgression to sampling (solid line in S4(A) Fig; see Methods), we find an excess of long tracts. The share of chromosome X coverage contained within the longest 1% of all tracts is greater than the share of chromosome X coverage within the remaining 99% of all tracts (S5 Fig). This pattern held for all additive models across all degrees of sex-bias. When variants were recessive and there were males among the introgressors ($p = 0$ and $p = 0.5$), this order was reversed.

## Coverage purging occurs on a fast timescale

In the preceding sections, we examined archaic coverage in extant haplotypes. We now consider the timecourse of archaic coverage retention from the introgression event to the sampling time.

Previous studies have observed that an equilibrium admixture quantity is achieved over a short timescale of tens of generations from the introgression event [7, 9, 20]. We recapitulate this result, and demonstrate its interaction with chromosomal inheritance type, dominance

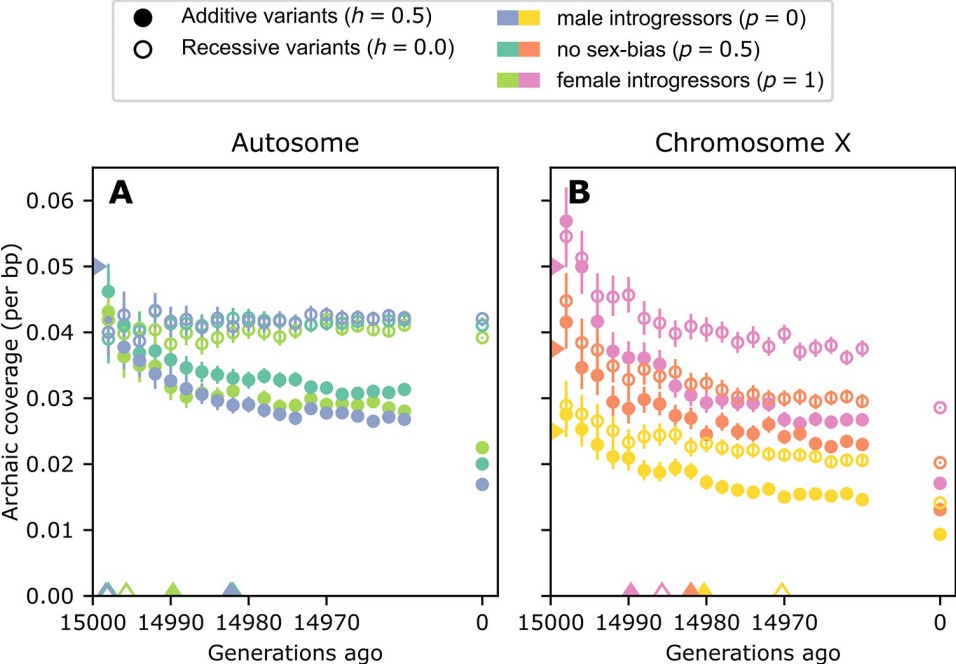

**Fig 4. The rapid timecourse of archaic coverage purging on the autosomes and chromosome X.** Carets on the x-axis indicate the generation at which mean coverage has fallen halfway to its final value from its value in the generation immediately following introgression. Carets on the y-axis indicate the initial introgression fraction. Error bars are bootstrapped empirical 95% confidence intervals around means of all samples across all simulation replicates. See S6 Fig for a visualization of the per-haplotype variance in archaic coverage over time. **A** Autosomal simulations. **B** Chromosome X simulations.

model, and sex-bias (Fig 4). We also observe a rapid temporal decay in the variance of archaic coverage across individuals (S6 Fig).

In all models, mean introgression coverage falls halfway from its value in the first generation after introgression to its final equilibrium value within the first 30 generations. However, when mutations are recessive, the timecourse of mean coverage in the autosomal model is remarkably stable (Fig 4A), while chromosome X still exhibits purging of recessive variants. These results are consistent with heterosis, and illustrate that selection coefficients inferred for autosomal recessive archaic variants do not translate to chromosome X [6, 7].

As expected, sex-bias does not influence the amount or degree of purging of autosomal coverage. On the other hand, on chromosome X the greatest differences in coverage amounts are driven by sex-bias ($p$, fraction of the introgressing individuals that are female). However, within 20 generations, $p$ no longer separates each curve, suggesting that sex-bias-related purging occurs on a shorter timescale than that due to the dominance model of the variants.

## Discussion

Studies of archaic coverage in extant human genomes have repeatedly identified considerably less Neanderthal and/or Denisovan coverage on chromosome X than on the autosomes (Figs 1 and 2, S1 Table). Several studies have attempted to explain this clear discrepancy by invoking hybrid incompatibility [1] or background selection [2, 6, 7]; the latter studies focused on background selection have posited that different distributions of selective effects on chromosome X and the autosomes explain observed data. Our study illustrates how sex-bias in demographic history, and specifically in the introgression event itself, can also generate lower archaic

coverage on chromosome X than the autosomes. Although sex-bias has been suggested as an explanation for the deficiency of archaic coverage on chromosome X [2, 6, 8, 21], to our knowledge this is the first study to directly investigate how sex-bias can shape archaic coverage. We demonstrate that varying the sex-bias of the introgressors alone can significantly alter the aut:chrX ratio of archaic coverage. For example, under a model of additive deleterious variants, this ratio is more than doubled between a scenario with female introgressors and one with male introgressors (Fig 3A, S2 Fig).

## Heterosis

We also demonstrate the relative effects of models of dominance and sex-bias on the dynamics of archaic coverage purging from the autosomes and chromosome X (Fig 4). The effects of dominance and rate of purging have been examined on autosomes, without consideration of sex-bias, by multiple previous studies of archaic introgression [7–9]. Our simulation studies show that the heterosis is strongest when maintaining recessive archaic variants on the autosomes (Fig 4). This is expected on the autosomes, as heterosis maintains archaic coverage in admixed individuals [7, 22] by increasing heterozygosity at sites previously homozygous recessive for a deleterious allele. While heterosis helps maintain high archaic coverage on autosomes across time, on chromosome X the archaic coverage drops over time. This makes sense, as the hemizygous chromosome X in males can not benefit from increased heterozygosity.

## Aut:ChrX coverage ratios in 1kG samples

We observe broad continental-level patterns in the aut:chrX ratio of archaic coverage among 1kG continental groups.

The EAS and EUR 1kG continental groups have respectively the least and greatest absolute coverage amounts on both autosomes and chromosome X, as well as differing aut:chrX coverage ratios (Fig 2). Several studies have examined differences between this pair of continental groups, and inferred distinct histories of archaic contact, including multiple introgression events [11, 23, 24]. The different timings and numbers of archaic introgression events between EAS and EUR likely contribute to why their aut:chrX coverage ratios and coverage amounts differ.

The high coverage ratios and high variance in archaic coverage found in AMR 1kG sample groups, particularly the Colombian (CLM), Mexican-American (MXL) and Puerto Rican (PUR) sample groups, may be due to recent admixture processes. In S7 Fig we provide a visualization of ADMIXTURE [25] results of the AMR samples presented in Fig 2, using supervised clustering with $K = 3$ and EUR, AFR, and combined EAS and SAS samples as reference groups. Recent admixture is evident in AMR 1kG sample groups, where African and Indigenous American ancestry is enriched on chromosome X in admixed individuals [26–28]. The AMR aut:chrX ratios therefore may display higher values than EAS and SAS due to the varying sources of their chromosome X coverage tracts.

In addition to modern admixture, continental groups have experienced different histories of archaic admixture [3, 4, 29] that may influence their aut:chrX coverage ratios. Continental groups represented by EAS and SAS contain introgressed DNA from Denisovan sources, whereas EUR 1kG continental groups broadly do not. Different magnitudes and degrees of sex-bias in any of these introgression events may contribute to the aut:chrX coverage ratio differences between EUR and EAS/SAS.

## Future directions

In this work, our simulations follow the demographic model used by Kim *et al.* [8] (S1 Fig), which generates autosomal archaic coverage amounts on the order of these observed in

humans today (S1 Table). We acknowledge that the true demographic histories are more complicated. Prior work on archaic introgression has mostly focused on quantifying archaic coverage, but we are only now starting to more fully characterize the introgression events: the timing of introgression, number of pulses, population size history, and how natural selection affects archaic variation in modern humans [11, 24].

Here, we simulate only deleterious variants, and these variants only accrue within exons. We also use the same variable recombination rate map for autosomal and chromosome X simulations (S4 Fig; although see S1 Appendix). However, proximity to genes [1, 7], as well as differences in the local exon structure and recombination rate between chromosome X and autosomes [30, 31] may contribute to differences in the local distribution of archaic coverage [6, 8, 22, 32]. Therefore studying the effect of drift and recombination on the quantity of archaic coverage that escapes background selection, and how these signals differ from positive selection around beneficial archaic variants [22] will require further investigation.

In addition, we have only focused on contrasting patterns of archaic introgression in the autosomes and the X chromosome. However, studies of non-recombining loci like the Y chromosome and the mitochondria have proposed that both the mitochondria and Y chromosome may have been replaced in late Neanderthals after an introgression from early modern humans into Neanderthals [33, 34]. Interestingly, in this scenario, no sex chromosome/autosomal differences are observed [3]. Modeling all chromosomes might provide insights into how the interplay of natural selection, sex-bias and different inheritance patterns result in distinct patterns of introgression.

Finally, we only consider parameters related to modern humans, but several studies have shown that introgression is common in other species [35–38]. For example, introgression from bonobos into the ancestors of central and eastern chimpanzees has been detected [36], and like humans, the amount of introgression is smaller on the X chromosomes. Another study detected introgression from a ghost lineage into bonobos, and introgression was also depleted on the X chromosome [37]. Our model structure can be adapted to other organisms to provide insights into speciation and how demography, genomic structure, sex-bias and natural selection have shaped the distribution of introgressed variants on the autosomes and chromosome X.

Our work has demonstrated that dominance and sex bias affect the evolution of archaic coverage on chromosome X because of its unique inheritance pattern. We have shown that the observed low level of archaic coverage on chromosome X could be explained merely by a reduction in the effect of heterosis and sex-biases in the introgression events, without involving a more complex model with hybrid incompatibilities. Our work also suggests that negative selection was likely acting on archaic variants, and provides an appropriate set of null models for evaluating positive selection on introgressed segments on chromosome X. In general, by incorporating sex chromosomes explicitly into modeling frameworks, our work leverages more information from genetic data and begins to outline the set of evolutionary scenarios underlying the history and evolution of introgressed genetic variation.

## Materials and methods

### Empirical coverage estimates

In Figs 1 and 2, we present published archaic coverage estimated from extant humans. References for the coverage estimates and data sources can be found in S2 Table. When necessary, we have converted all archaic coverage estimates to per-base pair rate of archaic genomic content, or the number of base pairs inferred to be of archaic origin, divided by either the total

length of chromosomes 1 through 22, or the length of chromosome X. Chromosome lengths were obtained from the GRCh37 assembly.

Fig 2 presents archaic coverage estimated by the method of Skov *et al.* [5] in data from the 1000 Genomes Project [18]. We apply `hmmix` to 1835 1kG samples genotyped on all of the autosomes and chromosome X. Data from 1831 individuals is presented; four individuals with autosomal to chromosome X coverage ratios greater than 50 or autosomal coverage rates greater than 0.1 have been removed.

All archaic coverage tracts were called using `-haploid` mode, which generates two output files for each individual: one for each haplotype. Chromosome X genotypes of males were made artificially-diploid. A per-base-pair rate was calculated for each individual by summing the lengths of all tracts across both haplotypes and diving by twice the length of either all the autosomes or chromosome X.

After calling tracts, we removed those with posterior probability <80%, and those that contained no SNPs matching the derived alleles from any of the high-coverage reference sequences provided by the `hmmix` package. From chromosome X, we removed tracts that overlapped with the pseudo autosomal region, and excluded the length of the pseudo autosomal region from our per-base-pair rate calculation.

Ratios of autosomal to chromosome X coverage are calculated within individual samples, not constructed by pairwise combinations of autosomal and chromosome X coverages (as in simulated data; see Simulation procedure).

All code necessary to reproduce Figs 1 and 2 is provided in the GitHub repository ramachandran-lab/archaic-chrX-sexbias.

## Simulation procedure

Forward genetic simulations under an explicit demographic model were implemented in the simulation software `SLiM v.3.3` [17]. Our simulation procedure and demographic model closely follow that of Kim *et al.* [8], although we have implemented sex-bias in the introgression pulse. For each simulation, we chose a model of chromosomal inheritance (autosomal or chromosome X), dominance ($h = 0.5$ or $h = 0.0$ for all mutations generated in the simulation), and introgressor sex-bias (fraction of introgressors that were female).

**Genomic model.** Our genomic model is identical to that described in the *Simulations of human genomic structure* section of Kim *et al.* [8], with the exception that our simulations used a scaling factor of 2, rather than 5.

Inheritance of chromosomes via either an autosomal or chromosome X pattern was managed by SLiM. Each simulated chromosome was 100Mb in length, with identical exon placement and recombination rate maps. The choice of exonic regions and variable recombination rate maps is described in [8]. See S1 Appendix for a sketch of S4 Fig results with different exon placement and recombination rate maps between chromosome X and autosomal simulations.

Chromosomes accumulated mutations within exonic regions. Mutations had selection coefficients drawn from a distribution fit to data from European individuals (EUR population, 1kG Phase 3 [18]; see [39]) of the form:

$$s \sim \Gamma(k = 0.186, \theta \approx 0.07068995).$$

**Demographic model.** We simulated a simple two-population model with a single-generation pulse of introgression from source to recipient (S1 Fig).

Simulations reflected a 25,000-individual recipient population, which received a pulse of introgression 5% of this size from a 250-individual source population after 20,000 generations

of isolation. Simulation was terminated 15,000 generations after the introgression pulse. Simulations were scaled down by a factor of 2 for computational efficiency, following best practices [17]. Following simulation in SLiM, a joint, pre-isolation history was appended to the simulation using the `recapitate()` method provided by `pyslim v.0.403`. All code necessary to perform the simulations is provided in the GitHub repository ramachandran-lab/archaic-chrX-sexbias.

**Introgression coverage calculations.** Once a simulation was complete, we calculated introgression coverage in each of 1000 haplotypes sampled from the final generation of individuals in the recipient population. A haplotype's coverage ($c_h$) is the proportion of its bases that are identical by descent (IBD) to one or more of the introgressing archaic haplotypes ($H^A$). Coverage is calculated as:

$$c_h = \frac{\sum_{t \in T_h} t_r - t_l}{L}, \tag{2}$$

where $L$ is the length of the haplotype in base pairs, and

$$T_h = \bigcup_{a \in H^A} \{[t_l, t_r], \cdots\}^a, \text{ with } t_r - t_l \leq 500 \tag{3}$$

where $t_l$, $t_r$ are the left and right endpoints of a tract of haplotype $h$ that is shared IBD with introgressing archaic haplotype $a$.

Haplotype sampling and IBD analysis for each simulation replicate were performed on the tree sequence data structure output from SLiM. The locations of the tracts shared IBD were obtained from the tree sequence using the `find_ibd()` method implemented in `tskit v.0.3.2` [40].

Due to limitations of the simulation software, simulations of autosomal inheritance and chromosome X inheritance were performed separately. Therefore, there is no natural pairing of particular autosomal and chromosome X haplotypes under which to calculate an aut:chrX coverage ratio. The simulated aut:chrX ratios presented in Fig 3A and 3B, S2 and S3 Figs is the mean and 95% confidence interval of 2000 bootstrap resamplings of the ratio of $n$ choices of autosomal coverages and $n$ choices of chromosome X coverage, chosen with replacement from all coverage data (which each contained at least $n$ unique haplotypes). For Fig 3, $n = 10,000$.

All code necessary to reproduce Figs 3 and 4 is provided in the GitHub repository ramachandran-lab/archaic-chrX-sexbias.

## Supporting information

**S1 Fig. Cartoon of demographic model implemented in simulations.** $N_r$ indicates size of recipient population in haplotypes; $N_s$ indicates size of source population in haplotypes; *kga* indicates thousand generations ago. Results presented here have $N_r = 100N_s$. Filled colors indicate generations in forward simulation; a prior coalescent history is appended to the ancestral population. See Methods for further details.
(TIF)

**S2 Fig. Aut:chrX coverage ratio decreases with greater female fraction of introgressors in simulations.** Chartreuse line depicts autosome to chromosome X haplotype ratio of a theoretical population with a given male fraction (Eq 1). Error bars are bootstrapped 95% confidence intervals around the ratio of 10,000 means generated from autosomal and chromosome X coverage distributions; in most cases error bars are smaller than the displayed points.
(TIF)

**S3 Fig. An unequal sex ratio in the recipient population interacts with introgressor sex-bias. A** Aut:ChrX coverage ratios from simulations that reflect both sex-bias within the introgression pulse, and a constant, ("standing") unequal sex ratio in the recipient population. Sex ratios shown are 25% female (standing male bias), 50% female (no standing sex-bias), and 75% female (standing female bias). Black points reflecting no standing sex-bias are equivalent to those shown in Fig 3A and 3C. Error bars are bootstrapped 95% confidence intervals around the ratio of 10,000 means generated from autosomal and chromosome X coverage distributions; in most cases the error bars are smaller than the displayed points. All variants are additive ($h = 0.5$). **B** Same as panel A, but all variants are recessive ($h = 0$).
(TIF)

**S4 Fig. Archaic coverage tracts on chromosome X are longer than those on an autosome in our simulation studies.** All results shown reflect an additive model of dominance. **A** Tract length spectra from simulations of an archaic introgression scenario under autosomal or chromosome X inheritance; note log scale. Points indicate frequencies of archaic coverage tracts within length bins of 3000 bp. Simulated chromosomes shared a variable recombination rate landscape (see Methods for details). Solid lines indicate expected archaic coverage lengths under a simple, neutral model with constant recombination rate equal to the mean rate of the simulated recombination landscape. **B** Average recombination rate within each simulated archaic coverage tract, plotted against length of the tract. Note log scale. **C** Proportion of archaic coverage tracts that are a given length or shorter; proportions are calculated within each inheritance type. Arrow indicates 95th percentile of tract lengths, illustrated in panel D alongside sex-biased scenarios. **D** Distribution of 95th percentiles of archaic coverage tract lengths found on either an autosome or chromosome X, across degrees of introgressor sex-bias. Each box reflects the length distributions from ten simulation replicates.
(TIF)

**S5 Fig. The longest archaic coverage tracts are preferentially found on chromosome X in our simulation studies.** Archaic coverage tract length spectra from a simulated autosome (blue) and chromosome X (orange). Each plot represents a combination of dominance ($h = 0$ or $h = 0.5$) and sex-bias ($p = 0$, $p = 0.5$, or $p = 1$, where $p$ is the female fraction of the introgressors). Pie charts depict fraction of total coverage found on the autosome or chromosome X. Right pie shows coverage contained within the longest 1% of tracts; left pie shows all other coverage. Note that chromosomes have identical size and local recombination rates; see Methods.
(TIF)

**S6 Fig. Variance in archaic coverage per haplotype decays rapidly after introgression.** Each point (note transparency) reflects the archaic coverage on one of 1000 haplotypes sampled from the recipient population every two generations for the first 40 generations after the introgression event. Crosses indicate mean coverage at each timepoint (see Fig 4). Horizontal line indicates initial introgression fraction.
(TIF)

**S7 Fig. Autosomal and chromosome X archaic coverage varies among AMR 1kG sample groups.** Colors indicate per-base pair archaic coverage estimates on autosomes or chromosome X, inferred by method of Skov *et al.* [5] using data from AMR samples from the 1000 Genomes Project [18], selected as described in Methods. Coordinates are ADMIXTURE [25] ancestry estimates for the individuals, using supervised clustering with K = 3 and EUR, AFR, and combined EAS and SAS individuals as reference groups.
(TIF)

**S8 Fig. Forward simulations of demographic processes can contain simulation error. A** Our simulations contain some stochasticity in both introgression proportion and introgressor sex ratio ($p$). Each point is an independent simulation run of the demographic model. The radius indicates the number of introgressing individuals as a proportion of recipient population (0.05 was simulated). The angle indicates the fraction of introgressors that were female ($p$ = 0, 0.2, 0.4, 0.5, 0.6, 0.8, 1 were simulated). The points plotted here for $p$ = 0, 0.5, 1 are the same simulations as those in Figs 3A, 3C and 4B. **B** Same as panel A, but from ten simulations at each $p$ using a revised simulation approach with no stochasticity in introgression proportion or sex-bias. Note that ten replicate points are all plotted at the same location. (Introgression fraction of 0.02 was simulated, rather than 0.05 as in panel A and the rest of the paper.) Panels **C**-**F** are equivalent to Fig 3, but use data from panel B. The mean ratios in panels C and D in this "neutral" scenario systematically underestimate the theoretical haplotype ratio expected for each sex-bias scenario.
(TIF)

**S1 Table. Genome-wide archaic coverage levels on autosomes and chromosome X inferred from human genomic data.** Coverage is the per-base pair proportion of the genome inferred to have come from archaic sources, found either on the combined 22 autosomes, or on chromosome X. The standard deviation across individuals is in parentheses. The fourth column shows the ratio of autosomal archaic coverage to chromosome X archaic coverage (Aut:chrX), which is always at least equal, and can range up to 18.8. All groups outside of Continental Africa have at least 3.3 times more relative coverage on autosomes than chromosome X.
(PDF)

**S2 Table. Methods used to estimate archaic coverage on chromosome X.** Coverage estimates in extant human groups are presented in Fig 1.
(PDF)

**S1 Appendix. Description of additional simulations with chromosome-specific exon and recombination maps.**
(PDF)

## Acknowledgments

We gratefully acknowledge Dr. Laurits Skov's assistance with his software `hmmix` [5], as well as productive discussions with present and former members of the Ramachandran and Huerta-Sánchez labs, particularly David Peede and Samuel Smith.

## Author Contributions

**Conceptualization:** Elizabeth T. Chevy, Emilia Huerta-Sánchez, Sohini Ramachandran.

**Funding acquisition:** Emilia Huerta-Sánchez, Sohini Ramachandran.

**Investigation:** Elizabeth T. Chevy.

**Methodology:** Elizabeth T. Chevy, Emilia Huerta-Sánchez, Sohini Ramachandran.

**Project administration:** Sohini Ramachandran.

**Resources:** Emilia Huerta-Sánchez.

**Software:** Elizabeth T. Chevy.

**Supervision:** Emilia Huerta-Sánchez, Sohini Ramachandran.

**Visualization:** Elizabeth T. Chevy.

**Writing – original draft:** Elizabeth T. Chevy.

**Writing – review & editing:** Elizabeth T. Chevy, Emilia Huerta-Sánchez, Sohini Ramachandran.

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
