## [Decision Letter · Decision Letter 0]

9 Nov 2022

Dear Dr Chevy,

Thank you very much for submitting your Research Article entitled 'Integrating sex-bias into studies of archaic admixture on chromosome X' to PLOS Genetics.

The manuscript was fully evaluated at the editorial level and by two independent peer reviewers. The reviewers appreciated the attention to an important problem, but raised some substantial concerns about the current manuscript. Based on the reviews, we will not be able to accept this version of the manuscript, but we would be willing to review a much-revised version. We cannot, of course, promise publication at that time.

We believe that the comments from the reviewers are generally highly relevant and even if they entail some reanalysis (particularly reviewer 2), this seems doable and could potentially increase the impact of the study.

If you decide to revise the manuscript for further consideration at PLOS Genetics, please aim to resubmit within the next 60 days, unless it will take extra time to address the concerns of the reviewers, in which case we would appreciate an expected resubmission date by email to plosgenetics@plos.org.

We are sorry that we cannot be more positive about your manuscript at this stage. Please do not hesitate to contact us if you have any concerns or questions.

Yours sincerely,

Mikkel H. Schierup

Academic Editor

PLOS Genetics

Bret Payseur

Section Editor

PLOS Genetics

Reviewer's Responses to Questions

**Comments to the Authors:**

Reviewer #1: In this manuscript, the authors explore the dearth of Neanderthal and Denisovan DNA on the X chromosome, compared to the autosomes. This pattern has been observed in several studies of archaic ancestry in modern humans, and the main point of this study is analysing how sex-biased introgression may have influenced this, by using simulations. This is a great example of using existing datasets and simulations for novel insights, and a manuscript where things are described with clarity and in a comprehensive manner.

I have only minor points which I suggest to address:

1) The first point concerns limitations of the methods to infer introgressed fragments. Such methods depend on mutation density – as a consequence, regions with low mutation rates will be more difficult to be identified as introgressed, even if they are. This is not a limitation of this study per se, rather of the underlying data for present-day humans in the initial observations. As I understood, heterogeneous mutation rates were not applied in the simulations (the link to the GitHub repo does not work yet), but since the “true” fragments are known, it does not matter for the result. However, this may be a factor in the real data leading to the hypothesis, because if the mutation rate of the X chromosome is lower, which I believe is the case, introgressed fragments would be less likely to be discovered. Most likely, it does not explain the whole pattern, but it might be useful to discuss this.

2) I found it a bit surprising that the absence of both archaic mitochondrial DNA and Y chromosomes in modern humans, with a likely replacement of both in Neanderthals (10.1126/science.abb6460, 10.1038/ncomms16046), was not mentioned in this context. Sex-biased migration, for example, with a male bias, would have increased the chances of transmission of the Y chromosome, but we don’t observe it.

3) On the other hand, gene flow from humans into Neanderthals seems to have been similar on the X chromosome and the autosomes (using ARGweaver, see citation [3] in the manuscript). This could be used to argue in favour of a sex-biased migration in the archaic-to-human scenario, however with a caveat of likely lower efficacy of selection in Neanderthals than modern humans (10.1073/pnas.1405138111).

4) The authors mention that introgression is common in other species (line 276). It might actually be useful to discuss what has been observed in the few species where this question was addressed: chimpanzees show a depletion of bonobo ancestry (measured with D-statistics; 10.1126/science.aag2602); bonobos show a depletion of “ghost” ancestry (8-fold compared to autosomes, using the Skov method; 10.1038/s41559-019-0919-x); the X chromosome seems not to differ from autosomes in introgression proportions in Southeast Asian macaques (10.1098/rspb.2021.1756); high-altitude pigs carry a long, positively selected haplotype that likely introgressed from an extinct species (10.1038/ng.3199). This kind of question is not completely unexplored in other species - especially our closest living relatives, which may serve as a direct comparison.

5) Somewhat related to the previous point, this might be added as a caveat: indeed sex-biased introgression might have contributed to what we see in humans, but given such similarities on the X chromosome of our closest living relatives (chimpanzees and bonobos), with very different sex-biased migration patterns and behaviours of their present-day populations, it is still an option that deleterious mutations are causing this. There is uncertainty of their past and our own past migration patterns, and the authors show that any such effect would have been immediate (Fig. 5) - so it seems very unlikely that this could be fully resolved. Here, I want to emphasize that this doesn’t impact the merit of the study, which lies in demonstrating the reasonable possibility of this factor. So, this last point is rather to think about the biology, and probably doesn’t need to be concretely addressed in the manuscript.

Reviewer #2: Summary

In this manuscript the authors are studying the remains of the admixture event between humans and Neanderthals. While the autosomes retain around 2% of Neanderthal ancestry the X chromosome has been depleted of archaic ancestry. The authors find that there is around 7.9 times more archaic sequence on the autosomes compared to the X chromosomes when studying individuals from the 1000 genomes project. They find differences between different human groups e.g. Americas has less archaic ancestry on the X chromosome compared to South Asian samples.

What is novel is that the authors look into the effect of sex-bias. They use simulations to argue that the results can be modeled by sex-bias (more neanderthal men than women). The manuscript is clearly written and its easy to follow.

Major comments

1.1) The authors find a Auto:X ratio of around 7.9 in the 1000 genomes data but their simulations only show a Auto:X ratio of 3 (Figure 3A,B). So there seems there are additional things at play which have much larger effects than the sex-bias.

The reduction in archaic ancestry on the X chromosome is a combination of:

dominance model (recessive or additive) + sex-bias + selective sweep + differences in recombination map + Hybrid incompatibility loci + Background selections + Demographics changes specific to different population

We appreciate the focus on sex-bias but in light of a recent preprint it would be good to pay extra attention to selective sweeps.

A recent preprint (Skov et al.)(https://www.biorxiv.org/content/10.1101/2022.09.19.508556v1.full)

found that there are 17 Mb of the X chromosomes in non-africans are affected by selective sweeps. These sweeps make up 10% of the X chromosome and effected haplotypes have almost no archaic admixture (mean = 0.0045%). The regions can be found in in Supplementary table S2 (start and end are in 100kb windows so a start of 195 means 19,500,000 bp).

This would presumably have an impact on the Auto:X ratio and make it appear higher than it actually is in 1000genomes data for instance. It would be good to repeat the analysis of the 1000 genomes data but excluding these regions.

1.2) Another issue is that when using hmmix (Skov et al 2018) one needs to change the parameters for the X chromosome. Since ¾ of mutations are passed on from men (Jónsson et al 2017) but the X chromosome only spends 1/3 of its time in men the mutation rate will be lower compared to the autosomes. It turns out that the emission parameters of hmmix needs to be multiplied by 5/6 on the X. This also changes the ratio for different 1000 genomes populations – see Supplementary table S3 for how it changes the ratios in the SGDP dataset (Skov et al.). We have also attached the relevant table with the calculated Auto:X ratios.

We feel that taking questions 1.1 and 1.2 into account will likely lower the Auto:X ratio to a point that is more compatible with the simulation studies – which would allow for some more concrete inferences about the human/neanderthal meeting.

1.3) The authors spend a lot of time discussing the difference in fragment length distribution of archaic segments on the X chromosome and the autosomes. (Figure 4) But as far as we can tell the results are not discussed and compared with real data – could the authors elaborate a bit on what we can learn from this analysis when comparing it to the 1000 genomes data? Also, It is not clear which new insights can be gained from studying the admixture proportion over time (Figure 5) that we didn’t already know from references (Harris and Nielen 2016, Kim et al 2018, Petr et al 2019) - perhaps the authors could elaborate a bit here?

1.4) We really liked Supplementary figure 2! Even more than the current main Figure 3. Maybe Supplementary figure 2 should be used in the main text instead of Figure 3 because it is really helpful to have the neutral scenario along side the scenarios where they simulated sex-bias and dominance model. It would also be helpful to see the neutral curve in Supplementary figure 2 simulated from the scenario described (Supplementary figure 1) instead of having the theoretical distribution. Also there is something strange going on with the confidence intervals - why is the ratio for the additive effect higher for a female fraction if 0.4 compared to 0.2 – it is strange that the confidence intervals are so narrow.

Minor comments

2.1) Fig 1. Would it be possible to add text to the diagonal lines indicating what the ratio is. For instance: Auto:X=1, Auto:X=2 and so on it would improve the readability of the figure. Perhaps you could also omit plotting the mean 7.9 as it is on top of the ratio=8 line (also because that ratio might be too high see 1.2).

2.2) Would it make more sense to have exchange Table 1 in the main text for SI Table 2 – this table have the Auto:X estimates which would be helpful to have in the main text instead of having to look it up in the supplement. You could still keep the population abbreviations of course.

2.3) In line 175 the authors state that:

“Panels A and B reflect data from simulations without introgressor sex-bias; panel C

illustrates that regardless of introgressor sex-bias, the 95th percentile of archaic coverage

tract lengths on chromosome X is longer than the 95th percentile of tract lengths on an

autosome”

Shouldn't this be panel D – as there is no sex-bias in panel C.

2.4) We would aviod using language such as “Differences among the coverage ratios are not cleanly correlated”. We would reserve words such as “significant” and “correlation” for when you are performing statistical tests. You could write Differences among the coverage ratios are not sorted by popultion”. In addition in Figure 4, panel A the Y-axis is labeled as frequency but it should be counts.

2.5 In line 75 you introduce the term aut:chrX but it is not defined. Could you add in in line 56 when you write: The autosomal to chromosome X coverage ratio (aut:chrX)

2.6) We are puzzled by Figure 5, Panel A. Why are does the archaic coverage change for the autosomes when you change the proportion of male introgressers – wouldn’t this always be the same?

**Have all data underlying the figures and results presented in the manuscript been provided?**

Reviewer #1: None

Reviewer #2: Yes

PLOS authors have the option to publish the peer review history of their article (what does this mean?). If published, this will include your full peer review and any attached files.

Reviewer #1: No

Reviewer #2: **Yes: **Laurits Skov

---

## [Decision Letter · Decision Letter 1]

16 May 2023

Dear Dr Chevy,

Thank you very much for submitting your Research Article entitled 'Integrating sex-bias into studies of archaic admixture on chromosome X' to PLOS Genetics.

The manuscript was fully evaluated at the editorial level and by independent peer reviewers. The reviewers appreciated the attention to an important topic but identified some concerns that we ask you address in a revised manuscript.

This is really a very minor revision where we just want you to have the chance to consider the remark from reviewer 2 in particular pertaining to some unexpected results.

We therefore ask you to modify the manuscript according to the review recommendations. Your revisions should address the specific points made by each reviewer.

Yours sincerely,

Mikkel H. Schierup

Academic Editor

PLOS Genetics

Bret Payseur

Section Editor

PLOS Genetics

Reviewer's Responses to Questions

**Comments to the Authors:**

Reviewer #1: The authors have fully addressed the minor points from my review. In my opinion, together with the improvements in response to reviewer 2, this is a great manuscript for publication in Plos Genetics.

Of note and related to my comment 4, meanwhile there is another study on archaic introgression in a close relative of humans, gorillas (10.1101/2022.12.19.521012). There, we also find a depletion of introgression on the X chromosome, so it seems indeed quite common.

Reviewer #2: The authors have responded to all my comments and I am happy to say that the manuscript has improved greatly - good job!

The simulation results are now much more similar to what is observed in 1000 genomes data. I think the simulation part of the study is solid so I only have major concern left.

When looking at Figure 2 (or Table 2 for that matter) It seems that Europeans have around 0.7% archaic coverage per base and East Asians have around 1.6% archaic coverage per base which is twice as much. For previous studies this difference is not as great (Skov et al 2018 finds 1.94% in EUR and you now find 0.91 in CEU for instance - I found the numbers in your Table 2) so I suspect something went wrong when running hmmix. If I was to take a guess I would say that you might be using individuals in your outgroup that have West Eurasian ancestry such as the 1000 genomes populations of LWK and GWD (see Figure 2 in https://www.nature.com/articles/nature15393). Which individuals did the authors include in the outgroup when running hmmix?

Once this issue is addressed I will have no further comments.

**Have all data underlying the figures and results presented in the manuscript been provided?**

Reviewer #1: None

Reviewer #2: Yes

PLOS authors have the option to publish the peer review history of their article (what does this mean?). If published, this will include your full peer review and any attached files.

Reviewer #1: No

Reviewer #2: **Yes: **Laurits Skov

---

## [Editor Report · Decision Letter 2]

10 Jul 2023

Dear Dr Chevy,

We are pleased to inform you that your manuscript entitled "Integrating sex-bias into studies of archaic admixture on chromosome X" has been editorially accepted for publication in PLOS Genetics. Congratulations!

Yours sincerely,

Mikkel H. Schierup

Academic Editor

PLOS Genetics

Bret Payseur

Section Editor

PLOS Genetics

Comments from the reviewers (if applicable):

**Data Deposition**

http://datadryad.org/submit?journalID=pgenetics&manu=PGENETICS-D-22-00990R2

**Press Queries**

---

## [Editor Report · Acceptance letter]

9 Aug 2023

PGENETICS-D-22-00990R2 

Integrating sex-bias into studies of archaic introgression on chromosome X 

Dear Dr Chevy, 

We are pleased to inform you that your manuscript entitled "Integrating sex-bias into studies of archaic introgression on chromosome X" has been formally accepted for publication in PLOS Genetics! Your manuscript is now with our production department and you will be notified of the publication date in due course.

With kind regards,

Dorothy Lannert

PLOS Genetics

On behalf of:
